# Exploring Patient Empowerment in Major Depressive Disorder: Correlations of Trust, Active Role in Shared Decision-Making, and Symptomatology in a Sample of Italian Patients

**DOI:** 10.3390/jcm13206282

**Published:** 2024-10-21

**Authors:** Alessandro Rodolico, Pierfelice Cutrufelli, Giuliana Maccarone, Gabriele Avincola, Carmen Concerto, Alfio Luca Cunsolo, Antonio Di Francesco, Rosaria Furnari, Ludovico Mineo, Federico Salerno, Vincenzo Scuto, Ilenia Tona, Antonino Petralia, Maria Salvina Signorelli

**Affiliations:** 1Department of Psychiatry and Psychotherapy, TUM School of Medicine and Health, Technical University of Munich, Klinikum Rechts der Isar, 80333 Munich, Germany; 2Psychiatry Unit, Department of Clinical and Experimental Medicine, University of Catania, 95124 Catania, Italy; giuliana.mcc@gmail.com (G.M.); avincolag@gmail.com (G.A.); carmenconcerto@hotmail.it (C.C.); cnsllc94l15c351a@studium.unict.it (A.L.C.); uni364709@studium.unict.it (A.D.F.); rfurnari@unict.it (R.F.); ludwig.mineo@gmail.com (L.M.); uni313841@studium.unict.it (F.S.); scuto.vincenzo@studium.unict.it (V.S.); tnolni89d70b429x@studium.unict.it (I.T.); petralia@unict.it (A.P.); maria.signorelli@unict.it (M.S.S.); 3Oasi Research Institute—IRCCS, Via Conte Ruggero 73, 94018 Troina, Italy

**Keywords:** patient empowerment, major depressive disorder, shared decision-making, trust in psychiatrists, patient active role in decision-making, mental health outcomes

## Abstract

**Background/Objectives**: Empowerment in medicine and psychiatry involves patients gaining control over health-related decisions, improving treatment adherence, outcomes, and satisfaction. This concept is especially significant in psychiatric care due to the complex challenges of mental health conditions, including stigma and impairment of emotional and cognitive functioning. We aim to investigate the correlations between patient trust, decision-making involvement, symptom severity, and perceived empowerment among individuals with Major Depression. **Methods**: Patients with Major Depressive Disorder were recruited in the “Policlinico G. Rodolico” psychiatry outpatient clinic from November 2022 to June 2023. Inclusion criteria: ages 18–65, ability to consent, stable condition, psychiatric medication history, and recent consultation. Exclusion criteria: psychotic features, bipolar disorder, substance abuse, high suicide risk, and severe comorbidities. Measures included the User Scale for Measuring Empowerment in Mental Health Services (SESM), Trust in Oncologist Scale (TiOS), Clinical Decision-Making Style for Patients (CDMS-P), and Hamilton Depression Rating Scale (HAM-D). Analysis used Kendall’s Tau correlation and Two-One-Sided Tests procedure. **Results**: Seventy-three patients completed the study. No relationship was found between decision-making involvement and perceived empowerment (τ = −0.0625; *p* = 0.448), or between trust in psychiatrists and empowerment (τ = 0.0747; *p* = 0.364). An inverse correlation existed between patient involvement in therapy management and trust (τ = −0.2505; *p* = 0.002). Depression severity inversely correlated with empowerment (τ = −0.2762; *p* = <.001), but not with trust or decision-making involvement. **Conclusions**: The lack of significant correlations suggests that decision-making involvement and trust alone may not suffice to enhance empowerment. Trust may encourage patient passivity, while skepticism might drive active involvement. Higher empowerment is associated with less depressive symptoms, highlighting its potential connection with patient outcomes.

## 1. Introduction

In medicine and psychiatry, empowerment refers to the process through which patients assume greater control over decisions and actions that impact their health [1]. It involves imparting the knowledge, skills, and self-assurance necessary for active participation in patient care. Empowerment is linked to improved treatment compliance, better health outcomes, and increased patient satisfaction [2]. By comprehending and participating in their care, patients are more likely to make informed decisions that align with their values and lifestyles, resulting in personalized and effective health care, a core point of the Shared Decision-Making (SDM) model [3]. In the psychiatric field, the principle of empowerment carries added weight. Mental health conditions often involve complex emotional, cognitive, and social challenges. Empowerment for patients with Major Depressive Disorder (MDD) involves enhancing their capacity to make choices and transform those choices into desired actions and objectives regarding their mental health and treatment. It involves patients developing the knowledge, skills, and confidence to actively participate in managing their condition. This concept encompasses making informed decisions about care, effectively communicating with healthcare providers, and taking an active role in recovery [4]. In the context of MDD, empowerment is particularly critical due to the inherent characteristics of the disorder, which commonly involve pervasive feelings of helplessness, diminished self-esteem, and cognitive impairments [5,6]. For individuals with depression, empowerment may include understanding their symptoms, recognizing triggers, and learning coping strategies. It also involves overcoming the feelings of helplessness often associated with depression, fostering a sense of self-efficacy, and actively engaging in treatment plans to improve overall mental health outcomes [7]. Consequently, empowering patients in psychiatric care requires instilling a sense of self-efficacy, fostering the development of coping strategies, and promoting autonomy, counterbalancing the feelings of helplessness and dependency that are commonly associated with mental health disorders [8]. Moreover, empowerment in psychiatry is associated with reduced stigma, as it advocates that individuals with mental health conditions can be recognized as active participants in their recovery rather than passive recipients of care [9]. The key factors identified in empowerment research for depressive disorders include self-efficacy, perceived control, health literacy, social support, and access to resources. Self-efficacy, defined as the belief in one’s capability to manage their condition, is positively associated with improved clinical outcomes. Perceived control influences engagement in treatment, while health literacy enhances the capacity for informed decision-making [10]. Robust social support networks are correlated with increased resilience and recovery rates [11,12,13]. Access to mental health resources, such as psychotherapy and pharmacotherapy, is critical for fostering empowerment, as these factors interact to shape patients’ experiences and their active involvement in treatment [14]. Patients with depression can be empowered by increasing their knowledge about their condition and treatment options, involving them in decisions that consider their preferences [15]. Healthcare providers can encourage active participation in therapy sessions and treatment planning, developing coping skills and self-management techniques that enhance patients’ ability to handle symptoms [16]. Fostering supportive relationships and promoting self-advocacy skills can boost confidence, providing access to resources and peer support groups, which allows patients to share experiences and learn from others [17,18]. Ultimately, empowerment involves helping patients recognize their strengths, set achievable goals, and take an active role in their recovery journey [19]. Recognizing depression as a clinical disorder rather than a personal shortcoming can enhance patients’ motivation to seek treatment and engage in therapeutic interventions; facilitating patients’ awareness of their symptoms and identifying precipitating factors can enhance self-monitoring and insight [20]. This increased awareness enables individuals to implement early intervention strategies upon recognizing prodromal signs, thereby mitigating symptom exacerbation [21]. Involving patients in shared decision-making regarding their treatment plan (e.g., pharmacological interventions, psychotherapeutic modalities) fosters a sense of agency and responsibility; collaborative care models, which integrate input from patients, healthcare providers, and mental health professionals, have been shown to improve treatment adherence and enhance patient engagement [22]. Additionally, the use of digital mental health tools, including mobile applications and online resources that provide symptom tracking, psychoeducation, and relaxation techniques, enables patients to independently manage their condition outside of clinical settings [23]. The limitations faced by patients with depression directly impact their empowerment potential; stigma and discrimination hinder patients’ ability to make autonomous decisions and actively participate in society [24], and limited access to quality mental health care and inadequate insurance coverage restrict patients’ choices and involvement in their treatment [25]. These barriers can diminish self-efficacy and the capacity to make informed health decisions [26]. By addressing these limitations, healthcare systems can enhance patient empowerment; improving access to care, reducing stigma, and ensuring equal rights can increase patients’ ability to take charge of their mental health, make informed choices, and actively engage in their recovery process [27].

The crucial importance of patient–physician factors in promoting empowerment has been highlighted, with trust in physicians and active patient participation in therapeutic decisions being key areas of focus in the literature [28]. Trust is a multifaceted aspect of the therapeutic relationship that encompasses various elements such as the psychiatrist’s perceived competence, empathy, and reliability [29]. The trust placed in physicians is often considered the cornerstone of effective patient care as it theoretically fosters an environment conducive to patient empowerment [4]. Several factors can inhibit patient trust in physicians, including ineffective communication, such as overly technical language or unclear explanations of diagnoses and treatments [30]. A lack of empathy, where the physician appears disinterested or disconnected from patient concerns, further diminishes trust [31]. Medical errors or inconsistencies in care, especially when poorly communicated, erode trust by suggesting incompetence or a lack of transparency [32]. Negative past healthcare experiences and cultural differences, including language barriers and misunderstandings of patient values, also contribute to reduced trust [33]. Trust is essential for patients to feel comfortable and secure in sharing their personal experiences and adhering to treatment recommendations [34]. However, the exact mechanism by which trust fosters empowerment remains underexplored. It is unclear whether trust directly leads to empowerment or acts as a facilitating factor within a broader set of influences [35]. The extent to which trust alone can facilitate empowerment is unclear and may be overestimated. The significance of the patients’ active involvement in therapeutic decisions cannot be overstated. This participation represents a shift from the traditional role of passive care recipients to active participants in the treatment process, collaborating with their psychiatrists. However, the extent to which this involvement empowers the patients remains uncertain [36,37]. It is crucial to determine if patients feel more in control and capable of managing their mental health when they are actively involved in decision-making or if some find the responsibility to be overwhelming [38]. The assumption that patient activity directly translates into increased empowerment without considering other mediating or moderating factors could be an oversimplification of a more complex dynamic. The purpose of this study is to explore the strength and direction of the correlations between patient empowerment and two crucial factors in psychiatric care, namely, trust in physicians and patients’ sense of participation in their own treatment, among individuals diagnosed with Major Depression. This study also assesses the relationship between these dimensions and the severity of depressive symptoms.

The research questions were as follows. 

Q1: Is there a correlation between patient decision-making involvement and their sense of empowerment? In which direction?

Q2: Is there a correlation between patients’ trust in psychiatrists and perceived empowerment? In which direction?

Q3: Is there a correlation between patient involvement in therapy management and trust level? In which direction?

Q4: What is the correlation between patient trust in the physician, decision-making involvement, feelings of empowerment, and depression symptom severity? In which direction? 

The significance of this research lies in its potential to inform more patient-centered approaches to MDD treatment, enhance the quality of patient–physician interactions, and ultimately contribute to better mental health outcomes. The study’s findings could potentially improve patient-centered care approaches, enhance treatment outcomes, and inform both clinical practice and future research in psychiatric care. By exploring these questions, the study aims to provide a more nuanced understanding of patient empowerment in psychiatric care, which could inform both clinical practice and future research in this field.

## 2. Materials and Methods

### 2.1. Recruitment

A cross-sectional observational study was conducted between November 2022 and June 2023. A total of 73 patients were recruited consecutively through convenience sampling in the outpatient services of the psychiatric unit of Policlinico “G. Rodolico” of Catania (Italy).

Before data collection, a preliminary interview was conducted to ensure that the inclusion criteria listed below were met. Diagnoses were made according to the criteria of the DSM-5-TR. Once the first phase was completed and eligibility for the study was confirmed, the patient was introduced to the study. This involved describing the procedures through a brief synopsis, answering the patient’s questions and then formally requesting informed consent. All the participants provided written informed consent. Participants did not receive any compensation for their participation.

The study was conducted in accordance with the Helsinki Declaration and approved by the Institutional Review Board (protocol number 2022/2).

Inclusion Criteria:Diagnosis of Major Depressive Disorder (MDD) according to DSM-5-TR criteria.Age between 18 and 65 years.Ability to provide informed consent.Stable medical condition for at least three months prior to study enrollment.History of psychiatric medication use for at least three months preceding study enrollment.Current engagement in psychiatric pharmacotherapy with consistent prescription management, evidenced by at least one psychiatric consultation within the past month.

Exclusion Criteria:Presence of psychotic features or diagnosis of psychotic disorder.Current diagnosis or history of bipolar disorder.Active substance abuse or dependence within the past six months.High suicide risk as assessed by clinical evaluation.Severe comorbid medical conditions that could interfere with study participation.

### 2.2. Scales Measures

#### 2.2.1. User Scale for Measuring Empowerment in Mental Health Services—Italian Version (SEMS)

The Italian version of the “User Scale for Measuring Empowerment in Mental Health Services” (Scala degli utenti per misurare l’Empowerment nei Servizi di Salute Mentale, SESM) [39] was designed to measure patient empowerment. This concept refers to the process that enhances individuals’ strength and their ability to actively control their lives and situation within a given social environment. The SESM consists of 28 items, each assessed on a four-point Likert scale ranging from ‘strongly agree’ to ‘strongly disagree’. The scale explores five domains: self-esteem and self-efficacy, ability and disability, community activism and autonomy, optimism and control over the future, and justified anger. It showed good reliability and internal consistency, with a Cronbach’s alpha of 0.86 in its Italian version validation [39]. The total score of the SESM ranges from a minimum of 28, indicating a low degree of empowerment, to a maximum of 112, signifying a high degree of empowerment. The selection of this scale is justified by its significant utility in mental health services for evaluating the level of empowerment among patients, a crucial component of mental health recovery and rehabilitation. The scale’s multidimensional framework facilitates a comprehensive assessment of the various dimensions of empowerment, which is vital for understanding and enhancing patient autonomy and their active involvement in the treatment and recovery process. By employing the Italian version of the scale, SEMS is tailored to the specific cultural and linguistic context of Italy, thereby enhancing the cultural relevance and accuracy of the measurements obtained.

#### 2.2.2. Trust in Oncologist Scale (TiOS)

The Trust in Oncologist Scale (TiOS) [40] is a self-administered scale that assesses a patient’s trust in their oncologist. This scale is a modification of the Patient Trust Scale (PTS), which itself is derived from the Trust in Physician Scale (TPS) [41], validated in three languages—Dutch (original), English, and Italian—following studies on oncology patients. Initially proposed by M. A. Hillen et al. in 2011, the TiOS consists of 18 items, each phrased in the third person singular, with response options ranging from “1” (strongly disagree) to “5” (completely agree). Three items (9, 11, and 13) are negatively worded, requiring reverse scoring (from “5” to “1”). The overall patient trust estimate is calculated by averaging the scores (range “1–5”) assigned to each item, with higher values indicating greater trust in the physician. The TiOS was chosen for our study on patients with depression primarily due to its status as the sole scale validated in Italian at the time our study commenced, and because of its exceptionally high internal reliability, since this scale surpasses others of its type in consistency, with a 0.88 Cronbach’s alpha [40]. Furthermore, we had access to its validated translation, and it represents a significant advancement over previous scales measuring trust, from which it was derived. The scale was chosen for its widespread clinical use and strong validation in Italian, unlike the Trust in Physician Scale (TPS), which lacks Italian validation. Our team has significant expertise with this scale, given our department’s close collaboration with the oncology service at the same institution, where we provide consultation and support for oncology patients. Additionally, the research team plans to use the TiOS in a separate oncology project, which is currently under evaluation, further enhancing our familiarity with its application and the potential for an extended dataset for further analysis. Using the TiOS in both oncology and psychiatry studies allows for future comparability of findings across different medical specialties, potentially yielding insights into trust dynamics in various healthcare contexts. This approach enhances the broader applicability and comparative value of the research findings, enabling cross-disciplinary comparisons and more comprehensive understanding of patient trust in different medical fields. Depression, recognized as a chronic condition, frequently encompasses a loss of hope and can be associated with death-related thoughts. The TiOS, with its comprehensive assessment of the patient–physician trust relationship, is particularly relevant in this context. Trust in healthcare providers is a critical factor in the management of chronic conditions like depression, where the therapeutic relationship can significantly influence patient outcomes. The dimensions of trust measured by the TiOS—Fidelity, Competence, Honesty, and Caring—are particularly pertinent for depressed patients. These dimensions encompass aspects like the perceived loyalty, expertise, truthfulness, and empathetic capacity of the oncologist, which are crucial for establishing a strong therapeutic alliance. This alliance is vital in the treatment of depression, where patient engagement and adherence to treatment plans are essential for effective management. The scale was chosen due to its extensive use in clinical settings and its robust validation in the Italian language.

#### 2.2.3. Clinical Decision-Making Style for Patients (CDMS-P)

The Clinical Decision-Making Style for Patients (CDMS-P) [42] comprises 21 items divided into three sections: Section A contains six items that explore patient preferences for autonomy in decision-making; Section B consists of three vignettes with nine items assessing decisional preferences; and Section C includes six items focused on the desire for information. Items in Sections A and C are evaluated using a five-point Likert scale ranging from ‘strongly disagree’ (0) to ‘strongly agree’ (4). Notably, Items 1-2-3-5-19 in these sections were scored inversely. In contrast, each item in Section B is rated on a scale from ‘patient’ (4) to ‘clinician’ (0). The scale’s validity was examined by dividing the sections into two subscales: Sections A and B form the Participation in Decision-Making (PD) scale, where higher scores indicate a greater desire for active patient involvement in the decision-making process. Section C corresponds to the Information (IN) scale, in which higher scores reflect a greater desire to receive information. The scale showed good reliability and internal consistency, with Cronbach’s alpha values remaining between 0.87 and 0.89, even when any single item was excluded [42]. The scale facilitates the identification of patients’ preferred decision-making styles regarding their healthcare. Understanding a patient’s decision-making style enables clinicians to tailor their communication strategies and therapeutic options to align more closely with the patient’s preferences. The application of the CDMS-P allows healthcare professionals to enhance communication with patients. For instance, some patients may favor a more direct and informational approach, whereas others might prefer a more collaborative and consultative model. By identifying a patient’s decision-making style, clinicians can adjust their recommendations and present information in a manner that is more comprehensible and relevant to the patient, thus potentially improving adherence to treatment.

#### 2.2.4. Hamilton Depression Rating Scale (HAM-D)

The Hamilton Depression Rating Scale (HAM-D) [43,44] is a clinician-administered scale used to assess depression in adult patients with a diagnosed depressive disorder. It quantitatively evaluates the severity of depressive symptoms and their changes after treatment. The original version was established by Hamilton in 1960 and comprises 17 items (HAM-D17), including depressed mood (I), feelings of guilt (II), suicide (III), initial insomnia (IV), middle insomnia (V), delayed insomnia (VI), work and interests (VII), psychomotor retardation (VIII), agitation (IX), psychic anxiety (X), somatic anxiety (XI), gastrointestinal somatic symptoms (XII), general somatic symptoms (XIII), genital symptoms (XIV), hypochondriasis (XV), weight loss (XVI), and insight (XVII). The latest version, which was used in this study, subdivides item XVI (weight loss) into two mutually exclusive subtypes (XVIa and XVIb), assessing subjective weight loss (reported by the patient or relatives) and objective weight loss (observed through repeated body weight measurements). Each variable is scored on a 5-point severity scale (0 = absent, 1 = mild, 2 = moderate, 3 = severe, 4 = very severe). The scale’s properties have been extensively validated and show robust internal consistency and inter-rater reliability [43,44].

### 2.3. Data Collection

Data were collected through patient interviews using paper forms containing questionnaires for self-administered measures, following the administration of informed consent. A preliminary interview was conducted prior to data collection to assess eligibility according to the inclusion and exclusion criteria and the depressive state using the Hamilton Depression Rating Scale (HAM-D); 73 were finally deemed suitable for the study. In cases of missing data in the self-administered questionnaire, patients were asked during the same session to fill in any gaps. This approach helped to avoid missing data and the need for data imputation.

### 2.4. Statistical Analysis

Kendall’s Tau correlation coefficient was utilized to determine the strength and direction of associations between two ranked variables. This choice was justified by the small size of our dataset, as Kendall’s Tau is more appropriate for smaller datasets due to its lower sensitivity to errors in the ranking when compared to Spearman’s rank correlation coefficient. Kendall’s Tau is particularly useful for assessing monotonic relationships, which is advantageous for analyzing non-normally distributed variables or in situations where the relationship is not linear [45]. To address categorical variables, we employed binary dummy variables. The interpretation of Kendall’s Tau ranges from −1 to +1, where values closer to ±1 indicate a stronger monotonic relationship. A Tau value between 0 and ±0.10 is considered very weak, between ±0.10 and ±0.20 is weak, between ±0.20 and ±0.30 is moderate, between ±0.30 and ±0.40 is strong, and above ±0.40 is very strong [46]. To evaluate the relevance and practical significance of our findings, we implemented the Two-One-Sided Tests (TOST) procedure. This method goes beyond traditional significance testing. It is specifically designed to determine whether or not an observed effect is statistically non-significant and falls within a predetermined range of equivalence. The TOST approach is particularly valuable when the goal is to demonstrate the absence of a clinically relevant effect rather than just the presence of statistical significance. For our study, we established equivalence bounds of −0.3–0.3. This range was selected based on the rationale that correlations below this absolute value probably are not clinically relevant, assuming they are only moderate [47]. By applying the TOST procedure, we aimed to confirm that non-significant results are not merely due to a lack of statistical power or sample size limitations but rather indicative of an actual absence of clinically relevant correlation. We set the significance threshold to *p* < 0.05 for initial analyses. To account for multiple comparisons, we applied Bonferroni correction, adjusting the *p*-value threshold accordingly (alpha = 0.05/6 = 0.008). Additionally, we calculated 95% confidence intervals (CIs). In adherence to “Standard 2.3”, as delineated by the American Educational Research Association [48], this study conducted reliability assessments for the scales and subscales utilized, to ensure the internal consistency of the data, with both Cronbach’s Alpha and McDonald’s Omega calculated and reported [49]. All statistical analyses were conducted using Jamovi (version 2.4), an open-source statistical software package based on “R” language, using the “psych” module [50,51]. Our sample size was justified by carefully considering the study’s exploratory nature, balancing statistical power with practical constraints. This approach aligns with current methodological thinking for observational studies, where rigid power calculations may be less applicable than reasoned judgment [52]. The chosen sample size aimed to provide sufficient data for meaningful insights while acknowledging the real-world limitations of participant recruitment and resource allocation [53].

## 3. Results

A total of 73 patients were successfully recruited for the study, and complete data measurements were collected from each participant.

Table 1 displays the demographic characteristics of our sample, which include gender, age, education level, employment status, marital status, living condition, mean duration of illness, and severity of symptoms. It also displays the mean and Standard Deviation (SD) for the scale measures, along with Cronbach’s alpha and McDonald’s Omega values, indicating satisfactory internal consistency for all assessments.

No significant correlation was found between Mean Duration of Illness, Education, Employment, Marital Status, and Living Condition and the variables of interest (CDSM, HAM-D, TiOS, SEMS); exceptions were made for Employment and SEMS scale (See Appendix A).

All patients were in a state of clinical stability, with the severity of symptoms ranging from mild to moderate, as assessed by the administration of the Hamilton Depression Rating Scale (HAM-D). Most participants were on complex medication regimens, typically combining an antidepressant with either a second-generation antipsychotic or a mood stabilizer as an adjunctive treatment. Some patients also received benzodiazepines to manage anxiety or aid sleep. Concurrently, approximately one-third of participants were undergoing psychotherapy. Appendix A presents comprehensive data on patients’ depression severity and the treatment regimens of each patient’s depressive symptoms, alongside detailed antidepressant information such as medication names, treatment duration, dosage, and quantity prescribed. The table also documents comorbidities, noting other chronic conditions affecting the patients. Furthermore, it outlines concurrent treatments, including both additional pharmacological interventions and non-pharmacological therapies.

Table 2 presents the identified correlations, detailing their τ, strength, and statistical significance. Additionally, it includes 95% confidence intervals and the results of TOST equivalence testing.

All findings that were not statistically significant exhibited correlation strengths that were equivalent to a negligible magnitude, as established by the Two-One-Sided Tests (TOST) procedure.

## 4. Discussion

The study found no significant correlation between patient decision-making involvement and their sense of empowerment, suggesting that patient participation alone may not enhance feelings of empowerment. Similarly, no link was observed between patients’ trust in psychiatrists and perceived empowerment, indicating that trust, while important, is not sufficient to foster empowerment. An inverse correlation was discovered between patient involvement in therapy management and trust levels, raising questions about balancing trust with active patient participation. Regarding depression symptom severity, no correlation was found with trust or decision-making involvement, but an inverse relationship was observed with empowerment. These findings challenge common assumptions about patient empowerment and involvement in mental health care, highlighting the complex interplay between trust, decision-making, empowerment, and depression symptoms. The absence of significant correlations between sociodemographic factors (duration of illness, education, sex, employment, living condition, and marital status) and our key variables suggests that empowerment challenges are widespread across all conditions and not confined to specific groups. This finding underscores the complexity of empowerment issues, indicating that each population segment likely faces unique obstacles. Similarly, concerns related to active participation and trust appear to transcend these demographic boundaries, highlighting the need for nuanced, context-specific approaches to address these issues effectively.

Here is what we found specifically in respect to our research questions: With respect to Q1 “Is there a correlation between patient decision-making involvement and their sense of empowerment? In which direction?”, our data suggest that there is no significant relationship between perceived empowerment and decision-making style within our sample, at least in a positive direction.

To the best of our knowledge, this study is the first in the existing literature to explore the correlation between patient empowerment and their preference for a particular decision-making style, whether paternalistic or shared decision-making. However, studies regarding the influence of shared decision-making on empowerment have been carried out; in particular, a 2016 systematic review and meta-analysis included 11 randomized control trials states that the use of SDM appears to have small beneficial effects on treatment-related empowerment indices [54].

Our finding implies that decision-making style alone may not be an adequate factor for enhancing perceived empowerment, making it plausible that other factors primarily promote self-efficacy and self-determination, indicating that an active patient role in isolation may not be sufficient for this objective. Our research did not conclusively identify an inverse correlation; however, the notion that a directive approach by healthcare providers might sometimes unexpectedly increase patient empowerment cannot be dismissed. This might be because clear instructions can minimize confusion, a strong provider presence can offer reassurance, trust in medical expertise can instill confidence, and simplifying complex decisions can make patients feel more capable of acting [55].

In addition, the lack of correlation could also indicate that the desire to actively participate in the decision-making process is not dependent on the patient’s empowerment and self-determination abilities. Active participation could, therefore, occur independently of self-awareness and be limited solely to the willingness to participate actively in the decision-making process, regardless of whether the final decision is made by the patient or the clinician.

A similar consideration could be made regarding Q2 “Is there a correlation between patients’ trust in psychiatrists and perceived empowerment? In which direction?”, which has an analogous negative answer. The lack of correlation between trust and empowerment in our data indicates that trust, despite its inherent value, is not sufficient to foster empowerment on its own. This suggests that, while building trust is intrinsically important, additional elements are likely necessary to achieve empowerment.

In contrast, with respect to Q3 “Is there a correlation between patient involvement in therapy management and trust level? In which direction?”, our data indicated an inverse correlation between an active role in therapy management and trust, being in line with similar findings in the literature [56]. This suggests that patients with higher trust may tend to adopt a more passive role (downside of trust), readily accepting medical advice, whereas those with lower trust might be more skeptical and, thus, take a more proactive stance in managing their therapy. This raises the question of how trust in the clinician can result in the patient relying totally on the doctor, setting aside his or her ability to discern what is most akin to his or her own values and wishes. The inverse correlation we found lays the groundwork for exploring how trust in the clinician can be transformed into an increase in the patient’s desire to be an active part of the decision-making process, thus increasing the levels of willingness to participate, which will only be guided by the clinician in whom the trust is placed but leading to a final decision based on the patient’s values.

Additionally, it is possible that patients who become more involved in their treatment may experience a decrease in trust towards their physician, potentially because of disillusionment or a departure from their initial idealized perception of the healthcare provider [57]. Considering that the significant relationship specifically pertains to the participation component, whereas the data on the information component are neither significant nor indicative of a correlation, this suggests that participation in the decision-making process may have a more pronounced impact on trust. It could be that active participation might either erode trust or be driven by an initial lack of trust rather than merely the act of seeking information. This finding implies that the level of trust is more closely tied to the degree of patient involvement in treatment decisions than to the amount of information that they search for.

Answering Q4 “What is the correlation between patient trust in the physician, decision-making involvement, and feelings of empowerment and depression symptom severity? In which direction?”, there is an absence of a correlation between depressive symptomatology and the investigated dimensions; an exception was made for empowerment, which is inversely correlated. This suggests that depressive symptoms do not affect the underlying dynamics of trust and decision-making; patients may demonstrate consistent patterns in decision-making styles and trust-building processes, regardless of depression severity [58,59]. Conversely, empowerment appears to be correlated with depressive symptomatology, meaning that an increase in empowerment could benefit patients; conversely, depressive symptoms could impede the empowerment process.

This study has several noteworthy limitations. Firstly, utilizing the Trust in Oncologist Scale (TiOS) for assessing trust in psychiatric contexts warrants scrutiny. Although the TiOS was initially developed for oncology settings, it was selected due to its superior psychometric properties and the lack of a validated Italian instrument specifically designed for psychiatric trust assessment. The decision to employ this scale was predicated on the hypothesis that significant parallels exist between oncological and psychiatric trust dynamics. While this assumption may have some validity, it requires further empirical validation.

Another limitation is the lack of detailed data on physical comorbidities and family history. Although our sample excluded severe comorbid medical conditions, we did not extract detailed information from clinical records. Additionally, family history data were often inaccurate or incomplete due to difficulties in collecting accurate information from patients or familiar with poor health literacy. Also, while we acknowledge the importance of family support in depression treatment, we did not collect comprehensive data on patients’ support systems beyond basic socio-economic information (education, employment, marital status, and living condition). The generalizability of the study’s findings presents another limitation, as the results may not be broadly applicable to diverse psychiatric populations or healthcare systems, given the specific cultural and institutional context of the research. The research involved 73 patients, which may restrict the generalizability of findings to broader populations. This sample size, while suitable for an exploratory study, limits the statistical power and ability to detect smaller effects. Future research with larger, more diverse samples would be beneficial to confirm and expand upon these results. The limited sample size also constrains the ability to perform more complex statistical analyses or subgroup comparisons, potentially overlooking important nuances in the relationships between trust, empowerment, and depression symptoms. Furthermore, the cross-sectional design of the study precludes the examination of trust development over time, potentially overlooking important temporal dynamics in the patient–psychiatrist relationship. The reliance on self-reported measures introduces the possibility of social desirability bias, particularly when assessing sensitive topics such as trust in healthcare providers. Lastly, the study’s focus on patient perspectives, while valuable, does not capture the potentially divergent views of psychiatrists or other healthcare professionals involved in patient care. These limitations collectively underscore the need for cautious interpretation of the findings and highlight avenues for future research in this domain. Subsequent studies should aim to address these constraints by developing and validating trust measures specific to psychiatric settings, employing longitudinal designs, incorporating multiple perspectives, and comprehensively examining potential confounding factors.

## 5. Conclusions

Our study revealed several unexpected relationships in patient care. Neither decision-making involvement nor trust in physicians correlated significantly with patient empowerment, suggesting that empowerment is more complex than simply giving patients more control or building trust. We found an inverse correlation between trust and active involvement, indicating that higher trust may lead to less patient engagement, or more involved patients may develop less trust. This challenges the assumption that trust always improves participation. Trust and decision-making showed no significant correlation with depression severity, indicating that depressed patients do not necessarily trust less or participate less in decisions. However, patient empowerment demonstrated a strong inverse correlation with depressive symptoms, suggesting that empowering patients may help alleviate depression, or that less depressed patients feel more empowered. These findings challenge conventional approaches to patient care and highlight the need for more nuanced strategies to enhance patient empowerment, particularly in mental health settings. The findings of this study suggest several practical applications for enhancing patient care in Major Depressive Disorder (MDD). Given that our results showed no significant correlation between decision-making involvement and perceived empowerment (τ = −0.0625; *p* = 0.448), psychiatrists should be cautious about assuming that shared decision-making alone will increase patient empowerment. Instead, they could implement structured tools that not only promote patient involvement but also explicitly address empowerment.

To address the inverse relationship we found between trust and active involvement (τ = −0.2505; *p* = 0.002), clinicians might adopt a nuanced approach. This could involve encouraging patient participation while simultaneously reinforcing their expertise, aiming to maintain trust without inadvertently promoting passivity.

The significant inverse correlation between empowerment and depressive symptoms (τ = −0.2762; *p* ≤ 0.001) suggests that empowerment-focused interventions could be valuable additions to standard treatment protocols for MDD. These interventions should target the specific domains measured by the SESM scale, such as self-esteem, self-efficacy, and optimism about the future.

Healthcare institutions could create more empowering environments by providing targeted patient education programs that focus on enhancing health literacy and self-management skills specific to MDD. Given that trust in physicians did not correlate significantly with empowerment (τ = 0.0747; *p* = 0.364), these programs should go beyond building trust to actively cultivate patient autonomy and self-efficacy.

These strategies aim to foster a healthcare culture that values patient autonomy and active participation in mental health treatment, while recognizing the complex interplay between trust, involvement, and empowerment revealed by our findings.

Further research is crucial to develop effective interventions that can improve patient outcomes.

## Figures and Tables

**Table 1 jcm-13-06282-t001:** Sociodemographic and clinical characteristics; Total Mean (SD) and internal reliability coefficients of our measures.

		N (SD)	%	
Mean Age		52.9 (16.5)		
Sex				
Male		32	43.8%	
Female		41	56.2%	
Education				
Elementary school		5	6.8%	
Middle school		32	43.8%	
High school diploma		27	37.0%	
University Degree		9	12.3%	
Employment status				
Not working/not studying		35	50.75%	
Retired		13	17.8%	
Studying		3	4.1%	
Working		22	30.1%	
Marital Status				
Single		22	30.1%	
Married		39	53.4%	
Separated/Divorced		8	11.0%	
Widowed		4	5.5%	
Mean Number of Children		1.1 (1.2)		
Living Condition				
Lives with family		58	79.5%	
Lives alone		15	20.5%	
Mean Duration of Illness		12.04 (9.57)		
Severity of symptoms	Symptoms free	Mild	Moderate	Severe
N (%)	15 (20.5%)	30 (41.1%)	28 (38.4%)	None
Scale		Mean (SD)	α	ω
CDMS		41.8 (9.06)	0.731	0.770
HAM-D		17.4 (10.34)	0.931	0.937
TiOS		70.7 (10.42)	0.908	0.933
SEMS		74.1 (8.57)	0.761	0.779

**Table 2 jcm-13-06282-t002:** Correlation between scales (Kendall τ).

					95% Confidence Interval			
		τ	Strength	*p*	Lower	Upper	Sig. Result	TOST Result
CDMS-P	HAM-D	0.0738	Negligible	0.369	−0.0819	0.2259	FALSE	TRUE
CDMS-P	TiOS	−0.2505	Moderate	0.002	−0.3901	−0.0997	TRUE	FALSE
CDMS-P	SEMS	−0.0625	Negligible	0.448	−0.2151	0.0932	FALSE	TRUE
TiOS	HAM-D	−0.0813	Negligible	0.323	−0.2331	0.0743	FALSE	TRUE
TiOS	SEMS	0.0747	Negligible	0.364	−0.0809	0.2268	FALSE	TRUE
SEMS	HAM-D	−0.2762	Moderate	<0.001	−0.4132	−0.1269	TRUE	FALSE

Hypothesis Tested: Equivalence; Bounds: −0.3–0.3.

## Data Availability

The data presented in this study are available on request from the corresponding author.

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
