# Peer review of "Exploring Patient Empowerment in Major Depressive Disorder: Correlations of Trust, Active Role in Shared Decision-Making, and Symptomatology in a Sample of Italian Patients"

_jcm, 2024, doi:10.3390/jcm13206282_

Round 1
Reviewer 1 Report
Comments and Suggestions for Authors
The language should be polished by native English speakers.
Introduction: The importance and significance of this study has not been fully described at present, perhaps the research on the factors related to empowerment and providing evidence for how psychiatrists can promote more empowerment is a good point to illustrate the importance of this study.
Methods: Why use the Trust Oncologist Scale (TiOS)(19) instead of the Trust Physician Scale (TPS).
Results: Table 1 and Table 2 can be combined into one, and the form format should be unified and standardized.
Educational and marital status were investigated, but the correlation and effect of the two were not explored.
Discussion: Education, gender, marital status were not discussed.
The limited sample size and lack of investigation into relevant factors should be mentioned in the limitation section.
Comments on the Quality of English Language
The language should be polished by native English speakers.
Author Response
Dear Reviewer,
Thank you for your thoughtful and constructive feedback on our manuscript. We appreciate the time and effort you've invested in reviewing our work. We have carefully considered each of your comments and have made substantial revisions to address them. Please find below our response to your concerns:
- Language polishing: We have had the manuscript thoroughly reviewed and edited by native English speakers to improve its clarity and readability.
If you would like to provide an example of remaining specific issues, we would be grateful for the feedback.
- Importance and significance: We have significantly expanded our introduction to better illustrate the importance of this study (lines 50-62; lines 153-160). We've added comprehensive information on factors related to empowerment in depression, including self-efficacy, perceived control, health literacy, social support, and access to resources (lines 68-77). We've also elaborated on how psychiatrists can promote empowerment, detailing strategies such as increasing patient knowledge, encouraging active participation, and fostering supportive relationships. (lines 78-108)
- Use of Trust Oncologist Scale (TiOS): We appreciate your query regarding our choice of scale. We've added an explanation in the Methods section detailing our reasons for using the TiOS instead of the Trust Physician Scale (TPS). Key factors include the TiOS's validation in Italian, our team's expertise with this scale, and the potential for cross-specialty comparisons in future research. (lines 236-248)
- Tables 1 and 2: As per your suggestion, we have combined these tables into a single, comprehensive table with a unified and standardized format.
- Educational and marital status correlation: We have conducted additional analyses on the correlation and effects of sex, educational and marital status. These results have been added as supplementary material and are referenced in the main text. (lines 367-371; lines 420-427)
- Discussion of education, sex, and marital status: We have expanded our Discussion section to address these factors. While our analysis found no significant correlations (see supplementary table), we have briefly discussed this lack of correlation and its implications, as it was not the primary focus of this article. (lines 420-427)
- Limitations: We have enhanced our limitations section to acknowledge the constraints of our sample size (73 patients) and its potential impact on the generalizability of our findings. We've also suggested directions for future research with larger, more diverse samples. (lines 519-525)
We believe these revisions have significantly strengthened our manuscript. We have strived to address all your concerns while maintaining the core focus and integrity of our research. We hope you find these changes satisfactory and look forward to your feedback.
Thank you again for your valuable input in improving our work.
Reviewer 2 Report
Comments and Suggestions for Authors
Unfortunately, the prevalence of major depressive disorder has increased significantly in the modern world. Often, drug therapy does not cope with this disease. It is very important for the patient to be committed to treatment and actively participate in it, this could significantly enhance the effect of depression treatment. Thus, the manuscript addresses a relevant topic. The manuscript under review is well structured and clear, the authors posed specific questions at the beginning of the manuscript, which are specifically answered at the end of the manuscript. However, the very concept of "empowerment" for patients with major depressive disorder requires clarification. What exactly do the authors mean by this concept? How do they propose to empower patients with depression? Are the rights and opportunities of patients with depression currently limited? If so, how is this expressed (references to studies or the authors' own research are necessary). Given that this part of the manuscript is not entirely clear, a question arises about the value of the entire study conducted by the authors. It is recommended to include answers to these questions in the "introduction" section. The authors should also expand the information on the observed group (section "Materials and Methods"): indicate the severity of the disease (the degree of depression), indicate what antidepressants the patients are taking (name, duration of administration, quantity and dosage); indicate concomitant treatment (were there other chronic diseases); whether the patients received non-drug treatment (hypnosis, meditation, etc.). All this will certainly affect the patient's involvement in the treatment process. It is also important to indicate the duration of the disease and family history. Therefore, it is recommended to add the above data to the tables with the results. Was the relationship between patient empowerment and the duration of the disease assessed? The results can be very interesting. Information on the support of relatives is also important. If these data are available, they should also be included in the manuscript. In the "conclusion" section, the authors should more precisely formulate the value of the obtained results and the possibilities of their practical application. How exactly can the obtained data influence the tactics of treatment of patients with major depressive disorder?
Author Response
Dear Reviewer,
Thank you for your thorough and insightful review of our manuscript. We appreciate your recognition of the topic's relevance and the structure of our paper. We have carefully considered your comments and have made substantial revisions to address them. Please find below our point-by-point response:
- Clarification of "empowerment": We agree that this concept requires further elaboration. We have expanded our introduction to provide a clear definition of empowerment in the context of major depressive disorder. We now explain that empowerment refers to the process of enhancing patients' ability to take control of their health and treatment decisions, increasing their self-efficacy, and promoting active participation in their care. (lines 50-62 and lines 78-108) Specific constraints and issues of patients with major depression are discussed. (lines 56-62 and lines 99-108)
We have added information on how we propose to empower patients, including:
- Increasing patients' knowledge about their condition and treatment options (lines 78-79)
- Encouraging active participation in therapy sessions and treatment planning (lines 80-81)
- Developing coping skills and self-management techniques (line 81-82)
- Fostering supportive relationships and promoting self-advocacy skills (lines 82-83)
- Providing access to resources and peer support groups (lines 83-84)
We have also included references to studies showing how empowerment can improve treatment outcomes in depression. (lines 68-77)
- Expansion of information on the observed group: We appreciate your suggestion to provide more detailed information about our study participants. We have updated the "Materials and Methods" section and relevant tables to include:
- Severity of depression (using standard HAM-D cuts off) (Table 1)
- Antidepressant medications (names, duration, dosage) (Supplementary Table 1)
- Non-pharmacological treatments (e.g., psychotherapy) (Supplementary Table 1)
- Duration of the depressive disorder (Table 1)
Regarding physical comorbidities and family history, we acknowledge that we do not have detailed data on these aspects (lines 509-511). Our sample was free of severe comorbid medical conditions, by exclusion criteria, but we did not extract this level of detail from the clinical records. We have added this limitation to our discussion section. Similarly, we found that family history data was often inaccurate or incomplete due to poor health literacy or understanding in our context. We have also noted this limitation in our discussion. (lines 511-513)
- Relationship between empowerment and disease duration: Following your suggestion, we have conducted additional analyses to assess the relationship between patient empowerment and the duration of the disease. We have included these results in our manuscript and discussed their implications, briefly since it is not the focus of our article. (Supplementary Table 2; lines 420-427)
- Family support: While we recognize the importance of family support in depression treatment, we did not collect comprehensive data on patients' support systems. We have acknowledged this as a limitation in our study and suggested it as an area for future research. (lines 513-516)
- Practical application of results: We have expanded our conclusion to more precisely articulate the value of our findings and their practical applications (552-574). Specifically, we now discuss:
- How understanding the relationship between trust, empowerment, and depression symptoms can inform treatment strategies (lines 552-557 and lines 562-566)
- Potential interventions to enhance patient empowerment and active participation in balance with trust in healthcare providers (lines 558-561)
- Implications for patient education and involvement in treatment planning (553-557)
- Suggestions for healthcare policy to support patient empowerment in mental health services (567-574)
We believe these revisions have significantly strengthened our manuscript and addressed the concerns you raised to the best of our ability given the constraints of our data. We have strived to provide a more comprehensive and nuanced presentation of our research while maintaining its core focus and being transparent about our limitations.
Thank you again for your valuable input in improving our work. We look forward to your feedback on these changes.